# Synthesis, Characterization, and Preliminary In Vitro Cytotoxic Evaluation of a Series of 2-Substituted Benzo [*d*] [1,3] Azoles

**DOI:** 10.3390/molecules26092780

**Published:** 2021-05-08

**Authors:** Ozvaldo Linares-Anaya, Alcives Avila-Sorrosa, Francisco Díaz-Cedillo, Luis Ángel Gil-Ruiz, José Correa-Basurto, Domingo Salazar-Mendoza, Adrian L. Orjuela, Jorge Alí-Torres, María Teresa Ramírez-Apan, David Morales-Morales

**Affiliations:** 1Instituto Politécnico Nacional, Departamento de Química Orgánica, Carpio y Plan de Ayala S/N, Escuela Nacional de Ciencias Biológicas, Colonia Santo Tomás, Ciudad de México 11340, Mexico; ozvaldo55@hotmail.com (O.L.-A.); fdiazc@ipn.mx (F.D.-C.); angeluzzy@hotmail.com (L.Á.G.-R.); 2Laboratorio de Diseño y Desarrollo de Nuevos Fármacos e Innovación Biotecnológica, Instituto Politécnico Nacional, Escuela Superior de Medicina, Ciudad de México 11340, Mexico; corrjose@gmail.com; 3Carretera a Acatlima, Huajuapan de León, Universidad Tecnológica de la Mixteca, Oaxaca 69000, Mexico; dsalazar@mixteco.utm.mx; 4Departamento de Química, Universidad Nacional de Colombia-Sede, Bogotá 111321, Colombia; alorjuelar@unal.edu.co (A.L.O.); jialit@unal.edu.co (J.A.-T.); 5Instituto de Química, Universidad Nacional Autónoma de México, Circuito Exterior, Ciudad Universitaria, Ciudad de México 04510, Mexico; mtrapan@yahoo.com.mx (M.T.R.-A.); damor@unam.mx (D.M.-M.)

**Keywords:** 2-substituted benzo [*d*] [1,3] azoles, in vitro cytotoxicity, breast cancer, ERα and GPER, molecular docking

## Abstract

A series of benzo [*d*] [1,3] azoles 2-substituted with benzyl- and allyl-sulfanyl groups were synthesized, and their cytotoxic activities were in vitro evaluated against a panel of six human cancer cell lines. The results showed that compounds BTA-1 and BMZ-2 have the best inhibitory effects, compound BMZ-2 being comparable in some cases with the reference drug tamoxifen and exhibiting a low cytotoxic effect against healthy cells. In silico molecular coupling studies at the tamoxifen binding site of ERα and GPER receptors revealed affinity and the possible mode of interaction of both compounds BTA-1 and BMZ-2.

## 1. Introduction

Heterocyclic structures are widely distributed in nature and occur as key fragments of biomolecules, e.g., amino acids, nucleic acids, vitamins, natural products, enzymes, biological targets, etc. Besides, they constitute numerous synthetic compounds with relevant applications in different branches of chemistry and technology [1,2,3,4].

The structures of benzo [*d*] [1,3] azoles, such as benzimidazole (BZM), benzothiazole (BTA), and benzoxazole (BOX), are highly valued heterocyclic scaffolds in the design and synthesis of new drugs and bioactive agents. These heterocyclic nuclei are privileged pharmacophores that exhibit an inherent affinity for various types of receptors involved in different biological signaling pathways, whose derivatives have shown multiple pharmacological functions, including anti-inflammatory, diuretic, antiviral, anti-insomnia, antiparasitic, anticancer, among others (Scheme 1) [5,6,7,8,9,10,11,12,13,14,15].

On the other hand, cancer is considered a complex group of diseases that can originate anywhere in the body, characterized by uncontrolled cell proliferation and abnormal growth of the affected tissue (neoplasm), being able to migrate causing metastasis. Cancer is the second leading cause of death worldwide; only in 2018 9.6 million deaths were estimated, with low- and middle-income countries being the most affected with approximately 70% of deaths. Lung, colorectal, stomach, liver, and breast cancers are metastatic diseases with the highest mortality rates [16]. Globally, breast cancer affects around 2.1 million women per year, accounting for approximately 15% of all cancer deaths in women [16,17]. Breast cancer is linked to estrogen receptors (ERs) called hormone receptor-positive (ER-positive) cancers. ERs are present in cells and are nuclear transcription factors with both dependent and independent transcriptional activity of the 17β-estradiol ligand, related to the physiology and pathophysiology of different tissues. The activated receptor is transported to the nucleus and binds to DNA to modulate the activities of different genes. Steroid hormones, including the ligand 7β-estradiol, play an important role in the development and physiology of the mammary glands, as well as being an important risk factor in breast cancer. These hormonal actions are mainly mediated by ERα and ER-β [18,19,20,21], which are strongly expressed in human breast cancer, ERβ being predominant in benign breast tumors, while ERα is dominant in malignant tumors [22,23,24], in such a way that ERα is associated with growth and cell proliferation in breast cancer [25,26]. Likewise, multiple studies have shown that G-protein coupled estrogen receptor 1 (GPER) is also involved in the transduction of estrogenic signals for the regulation of cell growth and proliferation in different types of solid tumors, such as colon cancers, lung, cervix, and breast cancer [27]. In fact, in approximately 50% of human breast cancers, the GPER has been found at high levels [28,29,30,31]. The genomic action mechanism of estrogens occurs: (1) when the ligand interacts with ERs, inducing a conformational change that activates them, or (2) a dimer is generated that is the active form, it translocates to the nucleus interacting with the estrogen response elements regulating the transcription of specific genes. In the case of GPER1, it is a component of the family of G-protein-coupled receptors, it is responsible for regulating cell growth (HER), including proliferation and apoptosis through a non-genomic response activating the epidermal growth factor [21]. Estrogen signaling is of great interest due to the high levels of ERs and GPER protein in breast cancers, for which these proteins are vital targets in the therapy of ER-linked breast cancer. Therefore, ERs signaling can be blocked, thus reducing proliferation. Tamoxifen (TAM) and other estrogen receptor selective drugs are used to treat or prevent breast cancer by inhibiting the proliferative effects of estrogen, mediated through ERs. The active metabolite of TAM, 4-hydroxy tamoxifen, exhibits a high binding affinity for ERs [32]. Among the main cancer treatments, chemotherapy using multiple antineoplastic drugs remains one of the most common basic alternatives. However, very often, these pharmacological treatments are accompanied by several serious side effects frequently not tolerated by the patients [33,34,35]. In this context, and considering the alarming statistics projected by the WHO for the next decade, there is a pressing urgency in the search for new targeted and effective chemotherapeutic agents for safer treatments. For this purpose, ERs are considered potential targets for the development of new anticancer agents.

A quick look at the literature shows several works on derivatives of BZM, BTA, and BOX, especially those substituted at the 2 position, which have shown important biological activity against different types of neoplasms, including breast cancer [36,37,38,39,40]. It has been pointed out that the mechanisms of action of these benzo [*d*] [1,3] azole are different, highlighting in an important way the signaling pathways modulated by ERs [41,42,43,44]. Thus, following our continuous interest in the synthesis of benzofused heterocyclic derivatives and the study of their potential biological activities, in this opportunity, we report here the synthesis and characterization of a series of benzo [*d*] [1,3] azoles substituted at position 2 with benzyl and allyl-sulfanyl groups (**BTA-1**, **BZM-2**, **BOX-3**, **BTA-4**, **BZM-5** and **BOX-6**) and their preliminary in vitro cytotoxic assays against different human cancer cell lines. Additionally, in silico molecular docking between the different heterocyclic derivatives (**BTA-1**, **BZM-2**, **BOX-3**, **BTA-4**, **BZM-5** and **BOX-6**) and ERα and GPER receptors where performed, seeking to explore the affinity and possible binding modes of these compounds with ERs. These molecular couplings were performed at the reference drug TAM binding site of both estrogen receptors.

## 2. Results and Discussion

### 2.1. Synthesis and Characterization

The series of 2-substituted benzo [*d*] [1,3] azoles (**BTA-1**, **BZM-2**, **BOX-3**, **BTA-4**, **BZM-5** and **BOX-6**) were easily synthesized by a procedure previously reported by our research group [45], involving the substitution reactions between 2-mercaptobenzimidazole and their analogs, 2-mercaptobenzothiazole and 2-mercaptobenzoxazole, with the corresponding 1-(bromomethyl)-4-vinylbenzene or allyl bromide. Most of the benzo[*d*]azoles heterocycle derivatives were obtained as pure microcrystalline solids in good yields (70 to 83%) with melting points between 49 and 140 °C (**BTA-4** and **BOX-6** were liquids) (Table 1).

The series of compounds were fully characterized by IR and NMR (^1^H, ^13^C{^1^H}) spectroscopies, mass spectrometry, and elemental analysis. Analysis by DART and EI-MS afforded spectra that display peaks with the molecular ions plus one mass unit (M^+^ + 1) for the compounds **BMZ-2**, **BTA-4**, **BZM-5**, and **BOX-6** analyzed by DART, while those analyzed by EI-MS (**BTA-1** and **BOX-3**) exhibited the expected molecular ion for the proposed structures. Additionally, analyses by ^1^H NMR of compounds **BZM-2** and **BZM-5** produced spectra exhibiting signals around *δ* 11 ppm due to the -NH- proton, while signals due to the heteroaromatic cores were observed between *δ* 8.0 and 7.0 ppm, where signals due to the hydrogens H4 and H7 are displaced to lower field at *δ* 7.91–7.42 ppm. Besides, signals due to the vinyl hydrogens appear in the region of *δ* 6.68–5.01 ppm, with multiplicities and couplings constants (*J*′s) typical for cis, trans, and geminal couplings. The methine signal (H13) of the benzyl-substituted derivatives **BTA-1**, **BZM-2**, and **BOX-3** is observed as a doublet of doublets (*dd*) with *J* = 17 and 10 Hz corresponding to the couplings with protons H14a and H14b, respectively. Conversely, in the allyl-substituted compounds **BTA-4**, **BZM-5**, and **BOX-6**, these signals were observed as multiplets due to the additional allylic couplings with proton H8. The latter signal for the benzyl-substituted compounds **BTA-1**, **BZM-2**, and **BOX-3** appear as singlets at *δ* 4.38–4.58 ppm, and for the allyl-substituted compounds **BTA-4**, **BZM-5**, and **BOX-6** as doublets (*J* = 6.9 Hz) at about 4.0 ppm. These spectroscopic data are summarized in Table 2.

Further analysis by ^13^C{^1^H} NMR yielded spectra displaying all the expected signals consistent with the proposed structures for all the compounds. In general, the signal due to the quaternary carbons attached to heteroatoms appears at lower field. While the signals due to the carbon C2 binding the three heteroatoms is located between *δ* 166.26 and 149.87 ppm, showing further deprotection for the BTA derivatives, and signals due to carbons C3a and C7a at the bicyclic linking bonds were observed in the range *δ* 153.22–139.27 ppm (see experimental section).

### 2.2. Preliminary In Vitro Antiproliferative Screening

With the compounds on hand, preliminary in vitro cytotoxicity studies were performed to explore their potential anticancer activity against a panel of six human cancer cell lines, i.e., CNS glia (U-251), prostate (PC-3), leukemia (K-562), colon (HTC-15), lung (SKLU-1), breast (MCF-7), and non-cancerous human gingival fibroblast cells (FGH) using TAM as reference drug. As shown in Table 3, the allyl-substituted compounds **BTA-4**, **BZM-5**, and **BOX-6**, did not cause cytotoxic effects against U-251, PC-3, and K-562 cancer cell lines, and they were also inactive against gingival fibroblast cells (FGH), while for the cell lines HTC-15, MCF-7, and SKLU-1, inhibition values ranging from 2 to 12% were observed. Besides, the benzyl-substituted compounds **BTA-1**, **BZM-2**, and **BOX-3** exhibited better growth inhibitory activity compared with those of TAM, being the best **BZM-2** and the worst **BOX-3**, being active only on three cell lines, i.e., HTC-15 (1.1%), MCF-7 (16.7%), and SKLU-1 (10.4%).

Interestingly, derivatives **BTA-1** and **BZM-2** displayed the highest cytotoxic activities of all heterocyclic derivatives against the six cancer cell lines explored. Thus, compound **BTA-1** showed moderate inhibition percentages of about 15% against U-251, PC-3, K-562, and HTC-15, being better than the allyl-substituted compounds **BTA-4**, **BZM-5**, and **BOX-6**. For breast cancer (MCF-7), derivative **BTA-1** exhibited moderate inhibition (21.4%) compared to TAM (71.3%), while the cytotoxic effect was better against SKLU-1 with 38.4% compared to 43.3% of inhibition for TAM. However, **BZM-2** was the one exhibiting the highest inhibitory profile among the series of benzo [*d*] [1,3] azole derivatives tested against the different cancer cell lines, being at least ten times more cytotoxic than the **BTA-1** derivative against the different types of carcinomas, thus indicating that the BZM core increases the cytotoxic activity compared with the BTA and BOX moieties. Hence, **BZM-2** exhibited the best cytotoxic effects against PC-3 (47.0%), K-562 (37.2%), MCF-7 (46.8%), and SKLU-1 (41%) cell lines. Notably, the inhibitory effect of compound **BZM-2** against PC-3 was 10% greater than the reference drug, and for SKLU-1, the inhibitory percentage is very similar, more notably the fact that it exhibited a very low inhibitory effect of only 7.7% in FGH non-cancerous cells, contrasted with the 100% observed for TAM. Regarding cytotoxicity in non-cancer cells, it can be observed that most of the heterocyclic derivatives have practically null or very low cytotoxicity, with **BZM-5** being the one with the greatest inhibitory effect (12%), still very low compared to TAM. In the case of the compounds with the best cytotoxic profiles against cancer cells from the in vitro studies, **BTA-1** and **BZM-2**, the cytotoxic effect on FGH is even lower compared to the reference drug, so that **BTA-1** is not active, and in the case of **BZM-2**, the cytotoxicity in healthy cells is very low, with a percentage of inhibition of about 8%. These values are positive and of great relevance in the search for more selective anticancer agents with low side effects. Finally, IC_50_ values for compound **BZM-2** were determined on PC3 (34.79 ± 0.13), K562 (22.79 ± 3.0), MCF7 (32.21 ± 0.78), and SKLU (27.93 ± 1.8).

Attempts to get IC_50_ values in SKLU with compound **BTA-1** revealed that the response was not concentration-dependent at μM concentrations of 100, 75, 50, and 25 where the r value for cytotoxicity was practically linear, and thus, if a value could be reported for this particular species, it would be >100 μM.

### 2.3. Molecular Docking Studies

In order to validate the performance of the molecular docking protocol, we used the crystalized X-ray ligand of the receptors with three docking programs. The receptors used for the molecular dynamic simulations were selected, since they are important targets in the development of several types of cancer. Thus, mammalian target of rapamycin receptor (mTOR) was selected, because it regulates relevant cellular pathways such as cell proliferation, autophagy, and apoptosis [46]. Additionally, the progesterone receptor (Pr), besides being a marker of the Er activity, its overexpression is observed in the development of breast cancer [47]. These facts fully justify the use of multitarget docking to evaluate the biological activity of the series of benzo [*d*] [1,3] azoles derivatives. The binding energy of these receptors were used as a reference to the in silico evaluation of the biological activity for the derivatives of benzo [*d*] [1,3] azole. To calculate the precision of the binding mode of the ligand, we calculated the root-mean-square deviation (RMSD) comparing the position of the ligand in the best poses obtained from the three programs and the crystalized ligand (Appendix A). We found a RMSD for all receptors lesser than 1 A° (Table 4), which is a good performance for docking simulations. In addition, the binding energy for the redocking process results in a strong biding energy with the three programs.

To select the best poses among all the conformations explored by the three programs, we used an exponential consensus rank approach. These results are shown in Table 5 in which it can be observed a good correlation between the experimental data and the molecular docking results. Since tamoxifen (TAM) was the crystal ligand in Erα; this was used as reference. As shown in Table 5, TAM has a strong interaction with all the receptors that could be related with an inhibition of the metabolic pathway. Compounds **BTA-4**, **BZM-5**, and **BOX-6** establish weak interaction with all the receptors, while compound **BZM-2** prefers the interactions with Erα and mTOR receptors. In addition, compound **BTA-1** interacts with the EGRF pathway and the ligand **BOX-3** with Pr. In spite of the molecular similarity, the series of compounds interact in different targets of cancer pathways.

To evaluate the ability of the derivatives of benzo [*d*] [1,3] azole to enter into the cellular membrane, we calculated the LogP for all the derivates. The results are presented in Table 5. As can be seen from this table, TAM and compound **BTA-1** presented the higher values of LogP compared with the other derivates. The sulfur atom in the azole group increases the hydrophobicity of the derivates **BTA-1** and **BTA-4** lowering the LogP values. From the whole group of ligands, the less hydrophobics were derivates **BZM-5** and **BOX-6**. These low values suggest a bigger difficulty to enter into cell and generate a pharmacological activity.

The derivatives of benzo [*d*] [1,3] azole exhibit the same pose and interactions with the receptor EGFR, since the size and charge in the atoms were similar, but the presence of an heteroatom changes the electronic structure of the azole derivate. All azole derivates form pi–alkyl interactions with Leu 718 and hydrogen bonds with the nitrogen atom from the azole group and the backbone of Met 793. The vinylbenzene group of the molecules **BTA-1**, **BZM-2**, and **BOX-3** exhibit a pi–sigma interaction with Thr 790 and pi–sulfur interaction with Met 766 as shown in Figure 1. The derivates **BTA-4**, **BZM-5**, and **BOX-6** establish weak interactions with the receptor, since the substituent group only forms a pi–alkyl interaction with Leu 844.

The Erα receptor presents interactions with all derivates through hydrogen bonds between the nitrogen atom from the benzo [*d*] [1,3] azole scaffold and Lys 449. Sulphur and oxo derivates **BTA-1**, **BOX-3**, **BTA-4**, and **BOX-6** form hydrogen bonds with Arg 394, meanwhile nitrogen atoms from derivates **BZM-2** and **BZM-5** form hydrogen bonds with Glu 353. All derivates establish pi–alkyl interactions with Pro 324. The vinylbenzene derivates **BTA-1**, **BZM-2**, and **BOX-3** exhibit pi–alkyl interactions due to their different electronic structure compared with the compounds **BTA-4**, **BZM-5**, and **BOX-6** (Figure 2).

All benzo [*d*] [1,3] azole derivatives present a pi–cation interaction with Arg 766 and hydrogen bond interactions between the nitrogen from the azole and Gln 725. The benzimidazole derivates **BZM-2** and **BZM-5** form hydrogen bonds with Asp 697. While the vinylbenzene derivates **BTA-1**, **BZM-2**, and **BOX-3** exhibit pi–sigma interactions with valine. The vinyl group in the derivates **BTA-4**, **BZM-5**, and **BOX-6** gives place to pi–alkyl interactions, as shown in Figure 3.

Compounds **BTA-1**, **BZM-2**, **BOX-3**, **BTA-4**, and **BOX-6** establish pi–pi stacking interaction with the benzo [*d*] [1,3] azole center and Trp 90, Tyr 57, and Phe 130. In addition, the nitrogen atom from the azole ring forms hydrogen bonds with the backbone of Tyr 113. While the nitrogen atom from compound **BZM-2** forms a hydrogen bond with Tyr 57, the benzo vinyl ligand exhibits a pi–pi interaction with Pro 120. Additionally, the vinyl group in derivates **BTA-4** and **BOX-6** does not interact with the receptor. The derivate **BZM-5** exhibits a different binding mode with two hydrogen bonds in the benzo [*d*] [1,3] azole ring (Figure 4).

As it can be observed, the molecular docking simulations described the stabilizing interactions between the derivatives and four proteins related to the development of various types of cancer. The in silico evaluation is in good agreement with the preliminary in vitro cytotoxicity bioassays. That is, compounds **BTA-1**, **BZM-2**, and **BOX-3** exhibit better interactions with the proteins due to the presence of the benzyl moiety in their structures thus facilitating the formation of stabilizing pi–sigma interactions. In contrast, the vinyl group in derivatives **BTA-4**, **BZM-5**, and **BOX-6** hampers the formation of any interaction with the proteins. Compounds **BTA-1**, **BZM-2**, and **BOX-3** also exhibited the larger values of LogP this being in favor of a better membrane transports of these species that in turn may also favor their cytotoxic activities, as was the case for compounds **BZM-2** and **BTA-1**.

## 3. Materials and Methods

### 3.1. Reagents and Apparatus

All reagents used were purchased commercially from Aldrich Chemical. Co. Inc. (St Louis, MO, USA), and used without further purification. Solvents were supplied by J.T. Baker (Phillipsburg, NJ, USA), which were dried and distilled prior to use, using standard procedures established under dinitrogen atmosphere. The melting points were determined (without corrections) using a MELT-TEMP II Laboratory Devices, vibrational spectroscopy IR were performed in the range of υ 4000 to 350 cm^−1^ in a NICOLETMAGNA spectrometer 750 FT-IR in KBr tablet. EI-MS and DART ± MS were carried out using a JEOL JMS-SX102A and JEOL JMS-T100LC spectrometer, respectively. NMR spectra were recorded in CDCl_3_ at room temperature on a JEOL spectrometer GX300 ECLIPSE at 300 MHz frequency for ^1^H and 75 MHz for ^13^C{^1^H}. The chemical shifts (*δ*) for ^1^H and ^13^C are reported in ppm down field of Si (Me)_4_ (*δ* = 0.0). The abbreviations used in the description of the NMR data are the following: *s*, singlet; *d*, doublet; *dd*, doublet of doublets; *ddd*, doublet of doublet of doublets; *td*, triplet of doublets; and *m,* multiplet. Coupling constants are reported as *J* in Hz. The number of protons (*n*) for a given resonance is indicated by *n*H. NMR spectra, mass spectra and docking results are available in Appendix A.

### 3.2. Synthesis of 2-Substituted Benzo[d] [1,3] Azole Heterocycle Derivatives (BTA-1, BZM-2, BOX-3, BTA-4, BZM-5, and BOX-6)

The series of 2-substituted benzo[*d*] [1,3]azoles derivatives were obtained by the following general procedure: To a suspension of 2-mercaptobenzothiazole (3.53 mmol) and K_2_CO_3_ (0.553 g, 4 mmol) in THF (30 mL), the corresponding 1-chloromethyl-4-vinylbenzene (or allyl bromide) (0.5 mL, 3.8 mmol) was added dropwise. The reaction mixture was stirred for 48 h at room temperature under N_2_ atmosphere. The reaction was monitored by thin-layer chromatography to completion. Then, the reaction was filtered and washed with CH_2_Cl_2_ (3 × 5 mL). The organic filtrate was dried over anhydrous Na_2_SO_4_ and evaporated. Solid compounds were recrystallized from hexane/AcOEt and dried under vacuum to afford the desired product.

2-(4-vinylbenzylthio) benzo [d] thiazole *(**BTA-1**)*

White solid (0.703 g, 2.48 mmol, 70%), mp 63–65 °C. IR (KBr), υ (cm^−1^): 3083, 3053, 3026, 2999, 2925, 2853, 1687, 1820, 1605, 1506, 1307, 1421, 1200, 1278, 1235, 1127, 1101, 1076, 992, 935, 905, 843, 750, 725, 673, 484, 434. EI-MS, m/z: 283 (70, [M] ^+^), 250 (30), 166 (10), 117 (100), 91 (15). ^1^H NMR (300 MHz, CDCl_3_), δ (ppm): 7.89 (ddd, J = 8.1, 1.2, 0.6 Hz, 1H, Ar), 7.73 (ddd, J = 7.9, 1.3, 0.6 Hz, 1H, Ar), 7.44–7.24 (m, 6H, Ar), 6.68 (dd, J = 17.6, 10.9 Hz, 1H, CH), 5.72 (dd, J = 17.6, 0.9 Hz, 1H, CH), 5.23 (dd, J = 11.0, 0.9 Hz, 1H, CH), 4.58 (s, 2H, CH_2_). ^13^C{^1^H} NMR (75 MHz, CDCl_3_), δ (ppm): 166.26, 153.14, 137.11, 136.27, 135.71, 135.33, 129.31, 126.49, 126.03, 124.27, 121.54, 120.98, 114.14, 37.40. Anal. Cal. for C_16_H_13_NS_2_ (283.41 g/mol): C, 68.08; H, 4.68; N, 4.94; S, 22.63. Found: C, 68.12; H, 4.71; N, 4.99; S, 22.54.

2-(4-vinylbenzylthio)-1H-benzo[d]imidazole *(**BZM-2**)*

White solid (0.832 g, 3.12 mmol, 83%), mp 136–138 °C. IR (KBr), υ (cm^−1^): 3064, 3053, 3020, 2957, 2924, 2853, 2789, 2697, 2609, 1623, 1586, 1503, 1436, 1404, 1343, 1295, 1268, 1230, 1194, 1115, 1012, 982, 909, 875, 844, 812, 741, 615, 596, 477, 430. DART^+^-MS, m/z: 267 (100, [M]^+^ + 1), 151 (35), 117 (15). ^1^H NMR (300 MHz, CDCl_3_), δ (ppm): 7.42 (dd, J = 5.7, 3.0 Hz, 2H, Ar), 7.12–7.08 (m, 6H, Ar), 6.51 (dd, J = 17.6, 10.9 Hz, 1H, CH), 5.57 (d, J = 17.6 Hz, 1H, CH), 5.10 (d, J = 10.9 Hz, 1H, CH), 4.38 (s, 2H, CH_2_). ^13^C{^1^H} NMR (75 MHz, CDCl_3_), δ (ppm): 150.05, 138.27, 137.03, 136.30, 136.11, 129.17, 126.57, 122.58, 114.34, 114.22, 137.36. Anal. Cal. for C_16_H_14_N_2_S (266.36 g/mol): C, 72.15; H, 5.30; N, 10.52; S, 12.04. Found: C, 71.96; H, 5.34; N, 10.43; S, 12.01.

2-(4-vinylbenzylthio) benzo [d] oxazole *(**BOX-3**)*

White solid (0.798 g, 2.98 mmol, 80%), mp 49–54 °C. IR (KBr), υ (cm^−1^): 3044, 2928, 1822, 1626, 1599, 1541, 1497, 1451, 1404, 1343, 1286, 1241, 1214, 1132, 1096, 990, 928, 908, 844, 803, 742, 623, 488, 425. EI-MS, m/z: 267 (45, [M] ^+^), 234 (10), 117 (100). ^1^H NMR (300 MHz, CDCl_3_), δ (ppm): 7.62 (d, J = 7.4 Hz, 1H, Ar), 7.43–734 (m, 5H, Ar), 7.30–7.20 (m, 2H, Ar), 6.68 (dd, J = 17.6, 10.9 Hz, 1H, CH), 5.73 (d, J = 17.6 Hz, 1H, CH), 5.24 (d, J = 10.9 Hz, 1H, CH), 4.54 (s, 2H, CH_2_). ^13^C{^1^H} NMR (75 MHz, CDCl_3_), δ (ppm): 164.54, 151.94, 141.96, 137.34, 136.30, 135.41, 129.40, 126.65, 124.42, 124.05, 118.55, 114.42, 110.01, 36.43. Anal. Cal. for C_16_H_13_NOS (267.35 g/mol): C, 71.88; H, 4.90; N, 5.24; S, 11.99. Found: C, 71.94; H, 4.94; N, 5.18; S, 11.95.

2-(allylthio) benzo [d] thiazole *(**BTA-4**)* [49]

Amber liquid (0.753 g, 3.63 mmol, 75%). IR (KBr), υ (cm^−1^): 3060, 2978, 2920, 1941, 1902, 1860, 1636, 1560, 1456, 1423, 1307, 1274, 1234, 1158, 1125, 1075, 990, 921, 859, 752, 724, 669, 591, 548, 525, 427. DART^+^-MS, m/z: 208 (100, [M]^+^ + 1). ^1^H NMR (300 MHz, CDCl_3_), δ (ppm): 7.80 (d, J = 8.1 Hz, 1H, Ar), 7.67 (d, J = 7.9 Hz, 1H, Ar), 7.36–7.31 (m, 1H, Ar), 7.24–7.18 (m, 1H, Ar), 6.01–5.58 (m, 1H, CH), 5.30 (dd, J = 16.9, 1.1 Hz, 1H, CH), 5.13 (d, J = 10.0 Hz, 1H, CH), 3.92 (d, J = 6.9 Hz, 2H, CH_2_). ^13^C{^1^H} NMR (75 MHz, CDCl_3_), δ (ppm): 166.20, 153.22, 135.34, 132.33, 126.06, 124.29, 121.60, 120.98, 119.19, 36.26. Anal. Cal. for C_10_H_9_NS_2_ (207.32 g/mol): C, 57.93; H, 4.38; N, 6.76; S, 30.93. Found: C, 57.85; H, 4.35; N, 6.74; S, 30.87.

2-(allylthio)-1H-benzo [d] imidazole *(**BZM-5**)* [49].

White solid (0.701 g, 3.68 mmol, 70%), mp 138–140 °C. IR (KBr), υ (cm^−1^): 3079, 3051, 2925, 2867, 2729, 1911, 1818, 1789, 1686, 1624, 1604, 1504, 1452, 1420, 1307, 1277, 1234, 1196, 1198, 1126, 1101, 1076, 991, 935, 905, 843, 750, 725, 673, 599, 484, 435. DART^+^-MS, m/z: 191 (100, [M] ^+^). ^1^H NMR (300 MHz, CDCl_3_), δ (ppm): 7.42 (d, J = 2.8 Hz, 2H, Ar), 7.08 (dd, J = 5.7, 2.9 Hz, 1H, Ar), 5.90 (td, J = 16.8, 8.1 Hz, 1H, CH), 5.18 (d, J = 16.9 Hz, 1H, CH), 5.01 (d, J = 9.9 Hz, 1H, CH), 3.87 (d, J = 6.8 Hz, 1H, CH_2_). ^13^C{^1^H} NMR (75 MHz, CDCl_3_), δ (ppm): 149.87, 139.59, 132.98, 121.87, 118.46, 114.14, 35.36. Anal. Cal. for C_10_H_10_N_2_S (190.26 g/mol): C, 63.13; H, 5.30; N, 14.72; S, 16.85. Found: C, 62.97; H, 5.32; N, 14.68; S, 16.76.

2-(allylthio) benzo [d] oxazole *(**BOX-6***) [50].

Amber liquid (0.768 g, 4.01 mmol, 77%). IR (KBr), υ (cm^-1^): 3082, 2979, 2928, 1638, 1599, 1498, 1451, 1424, 1402, 1337, 1281, 1236, 1213, 1118, 1129, 1094, 987, 923, 888, 860, 805, 740, 622, 598, 426. DART^+^-MS, m/z: 192 (100, [M] ^+^), 117 (32), 89 (18). ^1^H NMR (300 MHz, CDCl_3_), δ (ppm): 7.60 (d, J = 7.6 Hz, 1H, Ar), 7.43 (d, J = 7.9 Hz, 1H, Ar), 7.29–7.21 (m, 2H, Ar), 6.08–6.00 (m, 2H, Ar), 5.39 (d, J = 16.9 Hz, 1H, CH), 5.21 (d, J = 10.0 Hz, 1H, CH), 3.95 (d, J = 6.9 Hz, 2H, CH_2_). ^13^C{^1^H} NMR (75 MHz, CDCl_3_), δ (ppm): 164.39, 151.93, 141.97, 132.18, 124.37, 123.99, 119.38, 119.83, 118.54, 109.96, 34.96. Anal. Cal. for C_10_H_9_NOS (191.25 g/mol): C, 62.80; H, 4.74; N, 7.32; S, 16.77. Found: C, 62.79; H, 4.68; N, 7.31; S, 16.72.

### 3.3. Biological Evaluation

The carcinogenic cell lines were supplied by the National Cancer Institute (USA). Antiproliferative assays against different human cancer cell lines were determined using the protein-binding dye sulphorhodamine B (SRB) in a microculture assay to measure cell growth, as described in the protocols established by NCI1 [51]. Cell lines were cultured in RPMI-1640 medium supplemented with 10% fetal bovine serum, 2 mM L-glutamine, 10,000 units/mL of sodium penicillin G, 10,000 lg/mL of streptomycin sulfate, 25 μg/mL of amphotericin B (Gibco), and 1% non-essential amino acids (Gibco). They were kept at 37 °C in a humidified atmosphere with 5% CO_2_. The viability of the cells used in the experiments exceeds 95%, as determined with trypan blue.

The cytotoxic activity of benzo [*d*] [1,3] azoles derivatives (**BTA-1, BZM-2, BOX-3, BTA-4, BZM-5, and BOX-6**) against human cancer cell lines: glioblastoma (U-251), prostatic adenocarcinoma (PC-3), chronic myelogenous leukemia (K-562), colorectal adenocarcinoma (HTC-15), lung (SKLU-1), mammary adenocarcinoma (MCF-7), and non-cancer gingival fibroblast cells (FGH). The different carcinogenic cell lines were removed from tissue culture flasks by trypsin treatment and diluted with fresh medium. From these cell suspensions, 100 µL containing 5000–10,000 cells per well were pipetted into 96-well microtiter plates (Costar), and the material was incubated at 37 °C for 24 h in a 5% CO_2_ atmosphere. Subsequently, 100 μL of a solution of each well was added the compound obtained by diluting the reserves. The cultures were exposed for 48 h to the compound at 25 µM concentrations. After the incubation period, cells were fixed to the plastic substrate by adding 50 µL of cold 50% acidic aqueous trichloroacetic. The plates were incubated at 4 °C for 1 h, washed with tap water, and air-dried. Cells fixed with trichloroacetic acid were stained with the addition of 0.4% SRB. Free SRB solution was removed by washing with 1% aqueous acetic acid. The plates were then air-dried, and the bound dye was solubilized by the addition of tris without 10 mM base buffer (100 µL). The plates were placed and shaken for 10 min, and the absorption was determined at 515 nm using an ELISA plate reader (Bio-Tex Instruments).

### 3.4. Methodology and Computational Details

Electronic structure calculations were carried out by using the B3LYP density functional. The initial structure of ligands was drawn with Avogadro Software [52]. For all atoms, we used the Pople 6 – 31 + g basis set. All ligands were fully optimized in their geometries, and all stationary points were characterized as minima through frequency calculations. Solvent effects (water) were included with the SMD continuum method [53]. Atomic charges to be used in the molecular docking simulations were calculated with the NPA scheme. All calculations were carried out with gaussian16 suite of programs [54].

To improve the Docking outcomes and reduce the effects of force fields and system dependence, we used an exponential rack consensus (ECR) proposed by Palacio-Rodriguez et al. [55]. The ECR of each molecule was described using a consensus score *P(i)* corresponding to the sum of the exponential rank of *j* programs, by following the following equation:(1)$$P\left(i\right)=\frac{1}{\sigma }\underset{j}{\sum }\exp\left(-\frac{r_{i}^{j}}{\sigma }\right)$$
where *σ* is the expected value of the exponential distribution respect to the number of molecules *i*, and $r_{i}^{j}$ is the rank of molecule *i* predicted with the program *j*.

We used this consensus for three molecular docking programs. AutoDock 4 (AD4) [56], AutoDock Vina (Vina) [57], and Smina (vinardo scoring function) [58]. Current cancer therapies seek to target the hormone receptors. The receptors considered in this work were Erα, PR, EGFR, and mTOR. These tridimensional structures were taken from PDB codes 3ERT [59], 4OAR [60], 2J6M [61], and 4DRH [62]. These structures were used as a target in previous computational works related with anticancer activity [63,64,65]. The receptors files were prepared in the graphic interface Auto-Dock Tools 1.4.5 (ADT) [56] where the water molecules were removed, and all hydrogens atoms were properly added, and nonpolar hydrogen were merged to carbon atoms. The Gasteiger charges were added in the receptors to prepare the pdbqt files. The charges of the ligands were taken from the NPA population analysis based on density functional theory calculations. In all docking experiments, a grid box of size 60 × 60 × 60° A^3^ in X, Y, and Z coordinates with a spacing of 0.375° A was used. The ligand–receptor complexes were analyzed with LigPlot + [66] PyMol [67] and Maestro Schrodinger programs [68].

## 4. Conclusions

We have presented a small library of 2-substituted benzo[*d*] [1,3]azoles derivatives; these species can be synthesized in a simple and good yield manner. Preliminary in vitro cytotoxicity bioassays with the series of compounds **BTA-1**, **BZM-2**, **BOX-3**, **BTA-4**, **BZM-5**, and **BOX-6** has been complemented with in silico molecular docking studies, these results suggesting that molecules containing the aromatic vinyl moieties are anchoring better to ERα and GPER than those including the allylic fragments. While the higher affinity of **BTA-1** and **BZM-2** may be due to the atoms with a higher electron density that they present (S and N), compared to **BOX-3**, which includes a harder, more electronegative O in its structure. The major bioactivity in vitro of **BZM-2** and its affinity for ERs may be due to the additional interaction by hydrogen bonds that it establishes with the nitrogenous unit -NH-. The latter compound **BMZ-2** is especially interesting for future studies and structural modifications, since it was not only the most active compound of the series, but also the one with the lower activity against healthy cells. Hence, further studies may allow us to shed further light in the design of other species with enhanced activities aimed to develop new, specific, yet safer antineoplastic agents, related to the breast cancer estrogen receptors (ERα and GPER).

## Data Availability

Data sharing not applicable.

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
