# Peer review of "Synthesis, Characterization, and Preliminary In Vitro Cytotoxic Evaluation of a Series of 2-Substituted Benzo [d] [1,3] Azoles"

_molecules, 2021, doi:10.3390/molecules26092780_

Round 1
Reviewer 1 Report
The manuscript by Linares-Anaya et al describe the synthesis of six benzo[d][1,3]azoles and preliminary cytotoxic data.
The series is really spare (only six derivatives, and among these, three already reported in literature), moreover the compounds showed only poor cytotoxic activity (in most cases being not active at all).
In my opinion, the manuscript is not significant and original enough to be published on Molecules.
Author Response
Dear reviewer 1:
We thank very much for your comments, however we still believe that the results presented in this MS are worth publishing and that the results for such simple species may lead the way to further improvements in the future. Some of which we are currently investigating in our lab and on time will be disclosed.
Thanks a lot

Reviewer 2 Report
This manuscript reports the synthesis and cytotoxic evaluation of 2-Substituted Benzo[d][1,3]azoles. The work seems carefully done and the introduction, experiment design, results, and conclusions are clearly presented except for MS data. The manuscript needs corrections as follows.
(1) Line 89 (Table 1): '2-mercaptobenzo[d]azole' should be corrected to '2-Mercaptobenzo[d]azole'.
(2) Lines 92 and 99: The authors described the EIMS data for compounds 4-6 in Materials and Methods, so 'ESI' should be modified to 'EI'.
(3) Lines 110 and 116 (Table 2): 'H15' should be modified to 'H14'.
(4) Line 116 (Table 2): 'R1' should be modified to 'R'.
(5) Lines 117 (Table 2) and 202: 'Si(CH3)3' should be modified to 'Si(CH3)4'.
(6) Line 199: 'MS-ESI' should be modified to 'EIMS'.
(7) Lines 203-204: 'simple', 'double', 'double double', 'double double double', 'triplet doubles', and 'multiple' should be modified to 'singlet', 'doublet', 'doublet of doublets', 'doublet of doublet of doublets', 'triplet of doublets', and 'multiplet', respectively.
(8) Line 232: The EIMS of compound 2 showed [M+H]+ ion at m/z 267. The authors should explain the reason why the EIMS of compound 2 did not show a molecular ion at m/z 266.
(9) Lines 251, 260, 269: The EIMS of compounds 4 (Mw 207), 5 (Mw 190), and 6 (Mw 191), showed [M]+ ions at m/z208, 191, and 192, respectively. The authors should explain the difference between molecular weights of compounds 4-6and their molecular ions in EIMS.
Author Response
Response to Reviewer 2
We greatly appreciate your valuable comments to improve the MS under review.
Point 1: Line 89 (Table 1): '2-mercaptobenzo[d]azole' should be corrected to '2-Mercaptobenzo[d]azole'.
Point 2: Lines 92 and 99: The authors described the EIMS data for compounds 4-6 in Materials and Methods, so 'ESI' should be modified to 'EI'.
Point 3: Lines 110 and 116 (Table 2): 'H15' should be modified to 'H14'.
Point 4: Line 116 (Table 2): 'R1' should be modified to 'R'.
Point 5: Lines 117 (Table 2) and 202: 'Si(CH3)3' should be modified to 'Si(CH3)4'.
Point 6: Line 199: 'MS-ESI' should be modified to 'EIMS'.
Point 7: Lines 203-204: 'simple', 'double', 'double double', 'double double double', 'triplet doubles', and 'multiple' should be modified to 'singlet', 'doublet', 'doublet of doublets', 'doublet of doublet of doublets', 'triplet of doublets', and 'multiplet', respectively.
Point 8: Line 232: The EIMS of compound 2 showed [M+H]+ ion at m/z 267. The authors should explain the reason why the EIMS of compound 2 did not show a molecular ion at m/z 266.
Point 9: Lines 251, 260, 269: The EI-MS of compounds 4 (Mw 207), 5 (Mw 190), and 6 (Mw 191), showed [M]+ ions at m/z 208, 191, and 192, respectively. The authors should explain the difference between molecular weights of compounds 4-6 and their molecular ions in EIMS.
Responses:
Besides in order to properly reply to the queries and questions made by the reviewers the theoretical calculations were repeated as well as some of the biological cytotoxic assays. In this process the student Adrian L. Orjuela and Dr. Jorge Alí-Torres performed the new docking experiments and the M.Sc. María Teresa Ramírez-Apan did the biological assays. That is why these new names have been added to the list of authors in the corrected version of the MS.
We thank very much reviewer #2 for its thorough revision and suggestions that certainly have yielded a better MS. In this sense, all the modifications and mistakes have been corrected as requested by reviewer #2 and highlighted in yellow on the corrected version of the MS.
Regarding points 8 and 9, we do apologize, as we made a mistake and these compounds were analyzed, not by EI-MS, but by the DART+-MS technique (JEOL JMS-SX102A). As a result the M+1 can be observed. This has been fixed and highlighted in yellow on the corrected version of the MS.
The English grammar has been thoroughly revised, you will se that the MS is now more amenable for reading.
Reviewer 3 Report
The manuscript by Morales and co. described the synthesis and the preliminary evaluation of a series of benzo-1,3-azoles for their cytotoxic properties towards six cancer cell lines. Morevoer, the authors proposed ERs and GPERs as potential target of these compounds by studying their in silico interaction with the protein surface recognized by tamoxifen, here used as reference compound. The paper results in general well-written and organized in a logical way but too preliminary results and many gaps- in my opinion- are present in the current form.
- In the introduction, I suggest the authors to explain in a more detailed way the role in the signaling pathway of ERs exerted by the bezo-1,3-diazole compounds reported in literature in order to make their study more appealing.
- Taking into consideration that the structure of the reported compounds is quite simple and not new, I suggest the authors to susbstantially reduce paragraph 2.1 as the chemical characterization by NMR and IR is already reported in the SI and there is not any particular reason to explain in details the NMR spectra of the compounds in the text.
- The biological evaluation pf the compounds is the most important part of the present paper in my opinion or at least it should be. This aspect has to be expanded and more deeply investigated. First, I’d suggest reporting the cytotoxicity of the compounds in terms of IC50 values: this makes easier to make a comparison with other compounds reported in literature. I also suggest the authors to stress the fact their compounds are little or not cytotoxic on the non-cancerous FGH cell line if compared to tamoxifen. Re-name the compounds in a precise and univocal way by numbering or by labelling them (BTA-1, BZM-2, BOX3 and son on). I suggest to introduce more experiments to give a proper biological characterization of the azoles for example by measuring their logP, by exploring other possible targets than ERs, as established for tamoxifen such for example the effect on caspases or on the mitochondria transmembrane potential (see e.g. Apoptosi, 6, 469–477(2001); Free Radical Biology and Medicine 143 (2019) 510–521511). Their
- The molecular docking studies are definitely interesting, but it seems that any relevant conclusion could be deduced by the authors. For instance, is there any particular feature in the model interaction that could be claimed as responsible for the major activity of compound BZM-2? Docking studies with GPER with compounds 4,5,6 are missing in the text although present in figure 2.
- Some mistakes/minor revisions are necessary to be addressed:
- In the abstract correct BMZ-1 with the BMZ-2.
- Keywords such as cytotoxicity or cancer are too generic. Keywords are important in order to make your work easy to be found in databanks.
- Lines 43-44: please re-phrase this sentence.
- Line 73: I’d say benzyl not aryl- group as substituent.
- Lines 107-108 are a repetition of Lines 105-106.
- I suggest substituting Nh (no inhibition) with NA (not active).
- Lines 149: remove the full stop after “carcinomas”.
- Line 172: the authors refer to Ala 350 but I can’t see this residue in figure 1.
- Line 218: I suggest the authors to make the procedure more general by removing “compound 1 as a white solid” with “the desired product”.
- Line 276-279 and lines 287-290 are a repetition.
- In the Conclusions, I don’t think appropriate talking about “structural diversity” (line 319) for describing the structure of the compounds here presented but rather I’d talk of them as a small library of benzo-1,3-azole based compounds.
For all these reasons I recommend the publication in Molecules only after a major revision of the manuscript, with an eye on the biological characterization of the compounds.
Author Response
We greatly appreciate your valuable comments to improve the MS under review.
Point 1: In the introduction, I suggest the authors to explain in a more detailed way the role in the signaling pathway of ERs exerted by the bezo-1,3-diazole compounds reported in literature in order to make their study more appealing.
Point 2: Taking into consideration that the structure of the reported compounds is quite simple and not new, I suggest the authors to susbstantially reduce paragraph 2.1 as the chemical characterization by NMR and IR is already reported in the SI and there is not any particular reason to explain in details the NMR spectra of the compounds in the text.
Point 3: The biological evaluation pf the compounds is the most important part of the present paper in my opinion or at least it should be. This aspect has to be expanded and more deeply investigated. First, I’d suggest reporting the cytotoxicity of the compounds in terms of IC50 values: this makes easier to make a comparison with other compounds reported in literature. I also suggest the authors to stress the fact their compounds are little or not cytotoxic on the non-cancerous FGH cell line if compared to tamoxifen. Re-name the compounds in a precise and univocal way by numbering or by labelling them (BTA-1, BZM-2, BOX3 and son on). I suggest to introduce more experiments to give a proper biological characterization of the azoles for example by measuring their logP, by exploring other possible targets than ERs, as established for tamoxifen such for example the effect on caspases or on the mitochondria transmembrane potential (see e.g. Apoptosi, 6, 469–477(2001); Free Radical Biology and Medicine 143 (2019) 510–521511). Their
Point 4: The molecular docking studies are definitely interesting, but it seems that any relevant conclusion could be deduced by the authors. For instance, is there any particular feature in the model interaction that could be claimed as responsible for the major activity of compound BZM-2? Docking studies with GPER with compounds 4,5,6 are missing in the text although present in figure 2.
Point 5: Some mistakes/minor revisions are necessary to be addressed: In the abstract correct BMZ-1 with the BMZ-2. Keywords such as cytotoxicity or cancer are too generic. Keywords are important in order to make your work easy to be found in databanks. Lines 43-44: please re-phrase this sentence. Line 73: I’d say benzyl not aryl- group as substituent. Lines 107-108 are a repetition of Lines 105-106.
I suggest substituting Nh (no inhibition) with NA (not active). Lines 149: remove the full stop after “carcinomas”. Line 172: the authors refer to Ala 350 but I can’t see this residue in figure 1. Line 218: I suggest the authors to make the procedure more general by removing “compound 1 as a white solid” with “the desired product”. Line 276-279 and lines 287-290 are a repetition.
Responses:
Besides in order to properly reply to the queries and questions made by the reviewers the theoretical calculations were repeated as well as some of the biological cytotoxic assays. In this process the student Adrian L. Orjuela and Dr. Jorge Alí-Torres performed the new docking experiments and the M.Sc. María Teresa Ramírez-Apan did the biological assays. That is why these new names have been added to the list of authors in the corrected version of the MS.
1. Although the theme by itself is ample, we have added a few more lines and the corresponding references on the introduction (second paragraph, page 2, lines 48-56 and 63-76 and reference section reference 33 that describes in great detail the signaling pathway, all this included in the corrected version of the MS) that we believe stresses out in a proper manner the relevance of the signaling of this receptors on breast cancer and thus highlights the importance of the research presented in this MS.
2. Has been properly addressed as requested, reducing the description of the characterization for the presented compounds in order to make it more amenable for reading.
3. In this point, we have stressed out along the corrected version of the MS the fact that the species presented are little or nontoxic to healthy cells compared to tamoxifen. And as requested the labelling of the compounds has been changed to the suggested nomenclature. Regarding the suggestion to determine IC50 values, they have been obtained and included in the corrected version of the MS for the most active compound BZM-2 on 4 cancer cell lines, i.e. PC3, K562, MCF7 and SKLU. Attempts to get values in SKLU with compound BTA-1 were also attempted, however it was clear that the response was not concentration dependent at concentrations of 100, 75, 50 and 25 where the r value for cytotoxicity was practically linear and thus if a value could be reported for this particular species would be 100 micromolar (lines 191-196, page 7 of the corrected version of the MS).
Besides, and in order to comply with the request of deepen more on the biological activity of the species presented and the why they have the activities observed. We calculated the LogP values, the experimental determination unfortunately couldn’t be performed due to limitations caused by the current sanitarium contingency. But, the calculated values reveal similar lipophilicity of the most active compound BZM-2 with
the reference drug tamoxifen, results that is on line with the results observed, given the relation than values of LogP keep with, among other things, membrane transport of the compound and thus favoring the effect of a given species. These values and a brief analysis have been included in the corrected version of the MS (page 8, Table 5, lines 221-234) as requested by reviewer #3.
4. As you can see the docking experiments were re-done in order to have not just better graphics but more precise and valid results. Hence, although the discussion did not variate the figures are more tractive and descriptive, and the interactions described on the text can be clearly observed on the graphics. The calculations are now complete and discussed as requested by reviewer #3. And overall, the discussion has been nourished based on the better analysis of the results attained and the sharp comments and requests made by reviewer #3. We like it better and hope you are satisfied with these results that indeed make the MS more complete, interesting, and amenable for reading (docking studies pages 8 to 11 of the corrected version of the MS).
5. All the queries and corrections raised on this point have been properly attended and have been included on the corrected version of the MS.
The English grammar has been thoroughly revised, you will se that the MS is now more amenable for reading.
Round 2
Reviewer 1 Report
The manuscript has been improved according to the revisions proposed by the other reviewers, however in my opinion the study remains too limited to be published in Molecules
Author Response
Response to Reviewer 1
We thank very much for your comments, however we still believe that the results presented in this MS are worth publishing and that the results for such simple species may lead the way to further improvements in the future. Some of which we are currently investigating in our lab and on time will be disclosed.

Reviewer 3 Report
The manuscript has been deeply revised and its scientific soundness increased. The authors carefully followed the suggestions I'd made about the previous version introducing a more complete biological evaluation of the small library of compounds they presented. In particular, a deep revision of the molecular docking studies was carried out, letting the reader appreciate the possible target of such a class of compounds in exerting their bioactivity.
Anyway, before being published the paper needs some minor revisions:
- Tamoxifen is usually referred to as TAM in paragraph 2.3. The same should be done in paragraph 2.2
- In paragraph 2.3, the authors should add some details for explaining why they chose those four receptors among the possible others.
- Check the captions under the figures 2,3,4: they cannot all be referred to the binding interactions to receptor Erα.
- Check the font for the aminoacid residues. I think that Leu and not LEU is the correct one (Eur . J . Biochem . 138. 9-37 1984). The same goes for all the others the authors mentioned.
Author Response
Response to Reviewer 3
We thank very much the comments of reviewer #3, comments that have been addressed and the corresponding corrections included in the revised version “R2” of the MS.
Regarding point 1, we have changed the word “tamoxifen” for its abbreviation “TAM” along the MS, of course the abbreviation is defined earlier on the MS (page 2, line 71) for the better understanding and easy reading of the corrected version of the MS.
Regarding point 2, we have added a paragraph in the corrected version of the MS (page 7, lines 192-199) to briefly describe why we chose the four receptors used in this study, this includes the addition of two new references that have been included in the corresponding section of the corrected version of the MS (page 17, references 49 and 50, lines 601-604).
Regarding point 3, the captions for the figures 2 (page 9, lines 254-255), 3 (page 10, lines 264-265) and 4 (page 10, lines 276-277) have been corrected in the new version of the MS, we apologize for this mistake.
Finally point 4, we have corrected the nomenclature for the amino acid residues as requested and according to the reference suggested, this has also been corrected on figures 1, 2, 3 and 4 for coherence with the text in the corrected version of the MS.

Round 3
Reviewer 1 Report
I have already expressed my opinion about this manuscript.
Although the authors attest: “…the results for such simple species may lead the way to further improvements in the future. Some of which we are currently investigating in our lab and on time will be disclosed.” I believe that this manuscript itself is too poor and the authors should add the compounds they are developing and the data they are collecting, in order to obtain a more significant work.
Of course, this is only my opinion, the Editor has the last word.
Best regards.